# Diverse Localization Patterns of an R-Type Lectin in Marine Annelids

**DOI:** 10.3390/molecules26164799

**Published:** 2021-08-07

**Authors:** Sarkar M. Abe Kawsar, Imtiaj Hasan, Sultana Rajia, Yasuhiro Koide, Yuki Fujii, Ryuhei Hayashi, Masao Yamada, Yasuhiro Ozeki

**Affiliations:** 1Department of Chemistry, University of Chittagong, Chittagong 4331, Bangladesh; 2School of Science, Yokohama City University, 22-2, Seto, Kanazawa-ku, Yokohama 236-0027, Japan; rajia_bio@yahoo.com (S.R.); yasukoide04@yahoo.co.jp (Y.K.); s192111d@yokohama-cu.ac.jp (R.H.); yamada.mas.ug@yokohama-cu.ac.jp (M.Y.); 3Department of Biochemistry and Molecular Biology, University of Rajshahi, Rajshahi 6205, Bangladesh; hasanimtiaj@yahoo.co.uk; 4Center for Interdisciplinary Research, Varendra University, Rajshahi 6204, Bangladesh; 5School of Pharmaceutical Sciences, Nagasaki International University, 2825-7, Huis Ten Bosch-cho, Sasebo 859-3298, Nagasaki-ken, Japan; yfujii@niu.ac.jp

**Keywords:** acicula, annelid, epidermis, immunohistochemistry, lectin, nephridium, nerve cord, setae

## Abstract

Lectins facilitate cell–cell contact and are critical in many cellular processes. Studying lectins may help us understand the mechanisms underlying tissue regeneration. We investigated the localization of an R-type lectin in a marine annelid (*Perinereis* sp.) with remarkable tissue regeneration abilities. *Perinereis nuntia* lectin (PnL), a galactose-binding lectin with repeating Gln-X-Trp motifs, is derived from the ricin B-chain. An antiserum was raised against PnL to specifically detect a 32-kDa lectin in the crude extracts from homogenized lugworms. The antiserum detected PnL in the epidermis, setae, oblique muscle, acicula, nerve cord, and nephridium of the annelid. Some of these tissues and organs also produced Galactose (Gal) or *N*-acetylgalactosamine (GalNAc), which was detected by fluorescent-labeled plant lectin. These results indicated that the PnL was produced in the tissues originating from the endoderm, mesoderm, and ectoderm. Besides, the localizing pattern of PnL partially merged with the binding pattern of a fluorescent-labeled mushroom lectin that binds to Gal and GalNAc. It suggested that PnL co-localized with galactose-containing glycans in Annelid tissue; this might be the reason PnL needed to be extracted with haptenic sugar, such as d-galactose, in the buffer. Furthermore, we found that a fluorescein isothiocyanate-labeled Gal/GalNAc-binding mushroom lectin binding pattern in the annelid tissue overlapped with the localizing pattern of PnL. These findings suggest that lectin functions by interacting with Gal-containing glycoconjugates in the tissues.

## 1. Introduction

Lectins are carbohydrate-binding proteins critical in facilitating cell–cell contact, a fundamental cellular process. For example, in animals, lectins are a critical component of innate immunity. Lectins are also present in biological domains and viruses. They act as receptors for various glycans, consisting of chains of monosaccharides, that cover the surface of eukaryotic cells, the cell wall of bacteria and archaebacteria, and the capsid or envelope of viruses. Cellular processes such as growth, quality controls, differentiation, and cell death are regulated through the interaction between glycans and lectins. Thus, glycobiology, or the study of glycan, is important for tissue regeneration and tissue engineering research.

Lectin–glycan interactions are weak and reversible but quite specific. Although the origins of the structures of lectins remain unclear, they may have evolved in response to the development of glycan structures. Based on the primary structures of various families of lectins, around 30 structural folds have been indexed in the database (https://www.uniprot.org/database/DB-0231 accessed on 1 August 2021) [1].

A family of R-type lectin is produced by the animals of the phylum Annelida, of the superphylum Lophotrochozoa [2], which have strong regenerative abilities and adaptabilities to environmental changes such as chemicals, microorganisms, and climates. Each class of Oligochaeta (earthworms) [3], Hirudinea (leeches) [4], Polychaeta (lugworms) [5,6], and Echiura [7,8] in the phylum Annelida have been reported to produce R-type lectins. R-type lectins are derived from the ricin B-chain, a protein toxin consisting of AB subunits, in castor bean seeds. They usually have the three-leaf clover-like β-trefoil fold with Gln-X-Trp (Q-X-W) sequences in each subdomain (https://unilectin.eu/unilectin3D/search?fold=b-trefoil& accessed on 1 August 2021). The physiological function of the R-type lectins in annelids is still unclear. Studying the R-type lectins may provide insight into the glycobiology of tissue regeneration in annelids.

The biology of marine annelids has been extensively studied. Genus *Perinereis*, also known as lugworms, is a type of benthos that belongs to the class Polychaeta in the phylum Annelida. *Perinereis* sp. is widely found in the Suez Canal, Pacific Ocean, Indian Ocean, and Red Sea [9]. They inhabit under the mud bottom of brackish water and subsist on the organic matters in the sand. While it is well understood that lectins from different organisms diverge in structural properties and biological activities, the localization of each lectin in different tissues during development remains unclear.

Previously, we isolated PnL, a d-galactose-binding lectin of the R-type lectin family with an anti-bacterial activity, from *Perinereis* sp. [5,10]. It was also found that lugworms generate glycans in the tissue sections that produced galactose residues [11]. However, to the best of our knowledge, the localization of lectins in different tissues has not been surveyed. In this study, an antiserum was prepared against PnL by immunizing rabbits with a polypeptide of 14 amino acids derived from the primary structure of PnL. We applied the antiserum to the cross-sections of the annelids and found that PnL was specifically localized around the body wall, gut, spindles, and neurons. Our study suggests that PnL is co-localized with endogenous ligand glycans having Gal residues in some tissues.

## 2. Results

### 2.1. Detection of PnL in the Crude Extract of Lugworm Tissues

The anti-PnL antiserum against 14 amino acids sequence (Appendix A, K1, yellow sequence) detected a 32 kDa band in the crude extract by Western blotting (Figure 1A, column anti-PnL). Three polypeptide fragments derived from PnL were reanalyzed and found to be similar to the N-terminal side domain of the earthworm lectin EW29 that contained two R-type lectin domains in the polypeptide. We found that the partial primary structure of PnL shared similarities with R-type lectins, including two Q-X-Ws and one P-X-W sequence characteristic of the R-type lectin family.

### 2.2. Diversified Localization Pattern of PnL in Lugworm Tissues

The localization patterns of PnL varied in the tissues of *Perinereis* sp. PnL was produced in the dorsal and ventral epidermis (gray triangles; Figure 1(C1–C3). PnL was characteristically found in the setae and their bundles (white and meshed white, respectively, in Figure 1(C1)), aciculum, oblique muscle, and nephridium (purple triangles in Figure 1(B2)) of the lugworm. The nerve cords were also a characteristic point of lectin localization (orange triangles in Figure 1(C3)). A weak signal was detected in the intestinal mucosa (green triangles in Figure 1(C2,C3)).

The magnified views revealed that PnL was secreted from specific cells in the epidermis (Figure 2(A1)). Besides, PnL was localized at the origin of the setae at the acicula (Figure 2(A2)). It was characteristic that lectins such as PnL were enriched at the nerve cord and nephridium (Figure 2(A3,A4)). These patterns were similar to the localization of an endogenous galectin in the tissues of American bullfrogs [12]. Overall, R-type lectin was found in extracellular locations after secretion from specific cells at the epidermis, setae, oblique muscle, acicula, nerve cord, and nephridium (Figure 2B).

### 2.3. Co-Localization of the R-Type Lectin and Glycan Ligands

In a previous study [11], we observed d-galactose (Gal)/*N*-acetyl d-galactosamine (GalNAc)-containing glycans in the cross-sections of *Perinereis* sp. (Appendix A). Here, the localization pattern of the Gal/GalNAc-containing glycans was found to be almost identical to the distribution of PnL. The superimposition of fluorescein isothiocyanate (FITC)-labeled *Agaricus bisporus* agglutinin (ABA) lectin detected setae, acicula, nerve cord, nephridium, and the epidermis (Figure 3(A3,B3)). Alexa488-labeled anti-rabbit immunoglobulin G (IgG) also detected similar patterns (Figure 3(A2,B2)). The pattern of PnL and ABA-lectin binding overlapped in many tissues, including setae, acicula, and nerve cord. These results suggest that PnL is present at the same location as an endogenous ligand of a lectin (Figure 3(A4,B4)).

## 3. Discussion

R-type lectins are representative of lectin families, and they have a subunit molecular mass of around 15,000. From pharges to humans, lectin-encoding genes exist in all organisms. Although the R-type lectins in different organisms have low similarity in amino acid sequence, they usually have three Q-X-W motifs in a subunit and triple tandem repeat sequences consisting of 40–50 amino acids. Generally, an R-type lectin domain (or subunit) is combined with another R-type lectin domain (or subunit), an enzymatic domain, or a toxin domain (or subunit). Each lectin protein is well characterized biochemically. However, the localization patterns of lectins in different tissues in an organism have not been elucidated.

Our immunohistochemical study found that PnL, an R-type lectin in marine annelids, existed in the characteristic organs in lugworms. In addition, PnL was found to co-localize with lectin ligands and glycans in the tissues, similar to the localization of galectins in some vertebrate tissues [12,13]. Not many lectins derived from marine invertebrates were extracted from tissue homogenates, so they seem to exist apart from tissues [14,15,16]. However, these lectins are found to locate around characteristic tissues such as the guts, coelome, and spickles [17,18,19]. Together with these results, PnL provided more evidence that this Gal-binding lectin locates at the surface of various tissues of PnL in lugworm tissues and suggested binding with their endogenous ligands.

PnL is a 32-kDa polypeptide with a glycan-binding profile; it recognizes the d-galactose residues in the glycans of glycoproteins and glycolipids [5]. Two Q-X-W and a P-X-W sequence were found in the partial amino acid sequences of a cleaved PnL. Furthermore, the antiserum generated by immunizing rabbits with a PnL fragment could detect PnL in various organs such as the epidermis, setae, acicula, nerve cord, and nephridium in the annelid. These results indicated that PnL is produced in the tissues originating in the endoderm, mesoderm, and ectoderm. In addition, glycans with d-Gal or d-GalNAc residues at the reducing terminal were detected by FITC-labeled lectin staining at the same place as PnL. This observation may explain why PnL and other R-type lectins in annelids require haptenic sugar to separate from tissues when they are purified. These lectins bind to their endogenous ligands in the tissues; thus, the addition of haptenic sugar dissociates the lectins from their glycan ligands.

Our results also help delineate the physiological functions of R-type lectins in annelids. The R-type lectin domain is found in various kinds of proteins; it is present not only in lectins but also in enzymes and toxins [2]. For example, pierisin, a famous toxin in arthropods, consists of an adenosine diphosphate ribosylation enzyme and four R-type lectin domains [20]. In the case of sea cucumber (phylum Echinodermata), a pore-forming toxin having two R-type lectin domains results in fibrous multimerization to penetrate the cell membranes on erythrocytes [21] whereas its configuration was similar to the Vibrio toxin [22] in pathogenic bacteria.

In the case of bivalves, another group of animals of Lophotrochozoa, the R-type lectin SeviL, in the gills, binds to characteristic asialo-GM1glycans of glycosphingolipids [23]. These R-type lectins that have a β-trefoil fold structure increase their transcription according to the bacterial challenge test, suggesting that they have roles as innate immune molecules [24,25]. In addition to the characteristic localization of the bivalves R-type lectin, when the lectin was administrated to mammalian culture cells that express asialo-GM1 on the cell surface, the lectin bound to the cells and induced signal transduction of the cell growth regulation [23]. This suggests that even in the case of physiological condition, the lectin–glycan interaction may act for cell regulative roles.

In addition to the presence of R-type lectin, d-galactose-containing glycosphingolipids are also found in Oligochaeta [26]. Since the phylogenetic distance between Oligochaeta and Polychaeta is not far, glycans with similar structures would have been found in lugworms. These glycans will provide a clue to the physiological functions of R-type lectins in the Lophotrochozoa. Transcriptome analysis has been performed in *P**erinereis nuntia* [27,28]. However, the complete sequence of the R-type lectins in the species has not been delineated. Even so, R-type lectins are likely to present in many organs, bind to their endogenous glycan ligands, and play a role in innate immunity in response to diverse environmental cues. PnL was suggested to have consisted of only lectin domains, same as the earthworm R-type lectin. Besides, this lectin previously showed in vitro antibacterial potential [10]. In this way, R-type lectins are likely to present in many organs, bind to their endogenous glycan ligands, and play a role in innate immunity in response to diverse environmental cues.

It became evident that the lectin was produced by all three embryonic germ layers of *Perinereis* sp. and co-localized with galactose-possessing glycoconjugates in lugworm tissues. It will be of interest to isolate and identify the ligand glycoconjugates, such as glycosphingolipids and glycoproteins, of PnL in annelids to dissect the physiological function of the R-type lectins. Integrating the data from functional morphology, bioinformatics, and biochemical studies of marine invertebrates with environmental science studies will be necessary to achieve a comprehensive understanding of the roles of R-type lectins.

## 4. Materials and Methods

According to a previous report [5], partial amino acid sequences of PnL were shown by Edman degradation to appear as a new primary structure. In this study, these partial sequences were analyzed again using the basic local alignment search tool (https://blast.ncbi.nlm.nih.gov/Blast.cgi accessed on 1 August 2021) and the protein domain database Pfam (http://pfam.xfam.org/ accessed on 1 August 2021)

The antiserum against PnL was raised in rabbits (Sigma-Aldrich Japan, Tokyo, Japan). A total of 10 mg of antigen, derived from the peptide sequence of CYFWLAPNQLWFYA that appeared during the primary structure analysis [5], was injected 3 times during 50 days; antiserum was collected using saturated NH_3_SO_4_.

PnL was purified from marine annelid *Perinereis* sp. (a fishing shop in Yokohama City) [5]. The crude extract of marine annelid was prepared as follows; 5 grams (wet weight) of the lugworm were homogenized to a paste with a razor blade and mixed with 10 ml of Tris-buffered saline (TBS) (10 mM Tris(hydroxymethyl aminomethane)-HCl (pH 7.4) containing 150 mM NaCl). The homogenates were centrifuged at 27,500× *g* in 50 mL centrifuge tubes for 1 h at 4 °C with a Suprema 21 centrifuge equipped with an NA-4HS rotor (TOMY Co. Ltd., Tokyo, Japan). The pellets were homogenized again with 3 volumes (*w*/*v*) of TBS containing 50 mM galactose overnight at 4 °C. The homogenate was centrifuged at 27,500× *g* for 1 h at 4 °C, and the supernatant was dialyzed against TBS till free galactose was removed. The dialyzed crude supernatant was centrifuged again at 27,500× *g* for 1 h at 4 °C.

Crude annelid extract and purified PnL were separated using sodium dodecyl sulfate polyacrylamide gel electrophoresis (SDS-PAGE) [29] and electroblotted onto polyvinylidene fluoride membrane at the constant current of 2 mA/cm^2^ for 30 min using the EZFastBlot solution in a semi-dry system (AE-1460, ATTO Co Ltd, Tokyo, Japan) [30]. The blotted membrane was blocked with EzBlockChemi (AE-1475, ATTO Co Ltd) at room temperature, incubated with anti-PnL primary antibody rabbit serum (1:1000 dilution) and HRP-conjugated secondary goat anti-rabbit IgG antibody for 1 h each, and colored with the 3,3′,5,5′-tetramethylbenzidine solution EzWestBlueW (WSE-7140, ATTO Co Ltd) as per the manufacturer’s instructions.

The cross-sections of the annelids were prepared according to a previous report [11]. Lugworms were fixed in 4% paraformaldehyde with 0.1% glutaraldehyde and 0.1 M cacodylate (pH 7.4) at 4 °C for 2 h, dehydrated using an ethanol gradient, cleared in Lemonsol (Wako Chemicals, Osaka, Japan), embedded in Histoprep (Wako Chemicals, Osaka Japan), sliced into 4-μm sections on a microtome, and mounted on poly-l-lysine-coated slides. The sections were deparaffinized in xylene, rehydrated in an ethanol gradient, and treated with 0.45% hydrogen peroxide in methanol for 45 min at room temperature to inhibit the endogenous peroxide activity.

The deparaffinized slides were blocked overnight with 1% (*w/v*) bovine serum albumin containing TBS, incubated with anti-PnL antiserum (diluted 1:500 with TBS) and Cy3-labeled anti-rabbit mouse IgG for 1 h each, counterstained with FITC-labeled lectins (Cosmo Bio Ltd., Tokyo, Japan), and detected by a fluorescence microscope, KEYENCE BZ-9000 (KEYENCE Co., Osaka Japan; λex/em = 494/520 nm or 550/570 nm or both) [11].

## Figures and Tables

**Figure 1 molecules-26-04799-f001:**
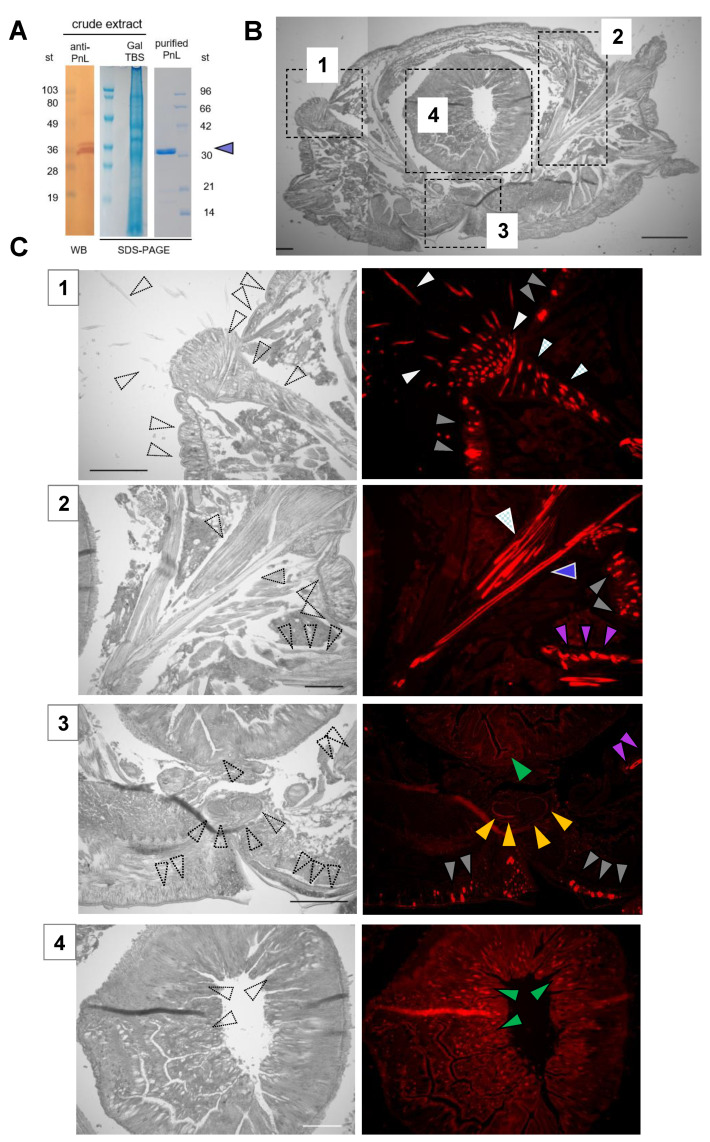
Localization of PnL in lugworm tissues. The PnL in the crude extract was detected by peroxidase staining of the HRP-conjugated goat anti-rabbit IgG raised against the PnL antibody (**A**). The bands in the crude extract and purified PnL (purple triangle) were also stained with Coomassie Brilliant Blue R-250. The numbers on the left and right are the molecular mass standards (st). Paraffin-embedded serial sections were observed using phase-contrast microscopy (**B**, (**C1**–**C4**), left). The sections were also immunostained with the antiserum against PnL and Alexa 488-conjugated secondary antibody, and observed using a fluorescence microscope KEYENCE BZ-9000 ((**C1**–**C4**), right; λex/em = 488/520 nm). Photos in **C** correspond to the areas in **B**. The epidermis (gray triangles), setae (white), bundle of chaeta (meshed white), acicula (blue), nerve cord (orange), nephridium (purple), and intestine (green) are denoted. Scale bars: 300 μm (**B**), 100 μm (**C**).

**Figure 2 molecules-26-04799-f002:**
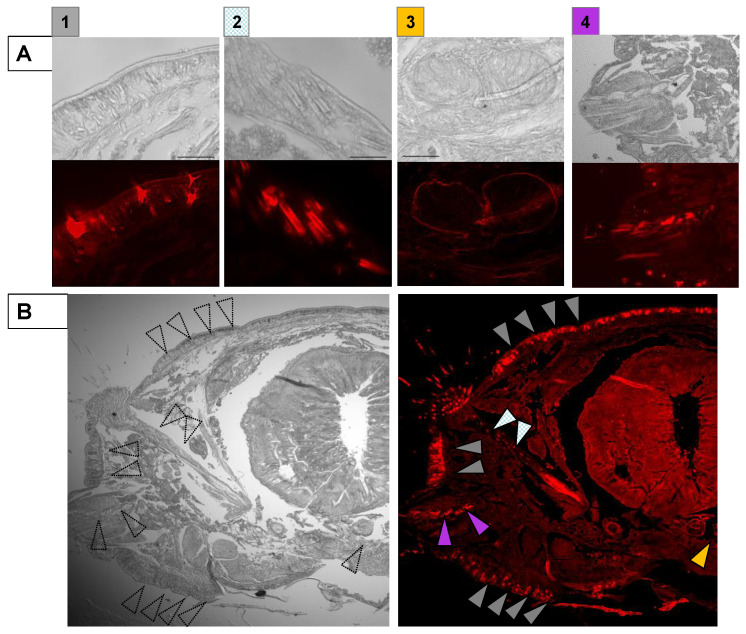
The tissues with PnL localization. (**A**) The epidermis (**1**, gray), acicula (**2**, meshed white), nerve cord (**3**, orange), and nephridium (**4**, purple). (**B**) The overview of the tissues. The colors of the arrows correspond to those in A. Scale bar: 50 μm.

**Figure 3 molecules-26-04799-f003:**
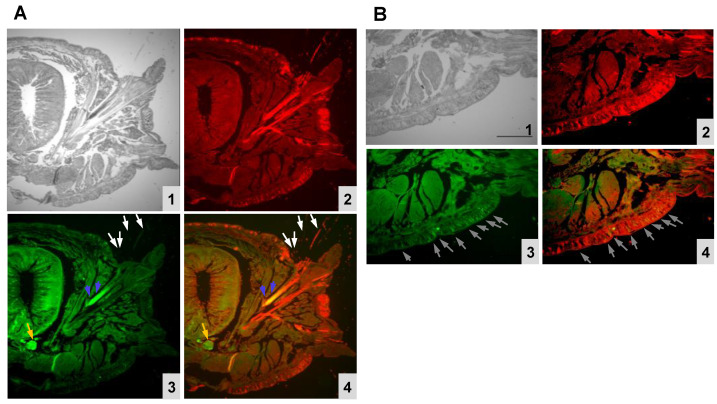
The localization patterns of PnL and *Agaricus bisporus* agglutinin (ABA)-lectin bindings. The *Perinereis* sp. tissues were 40× (**A**) or 80× (**B**) magnified, visualized with phase-contrast microscopy (**1**), anti-PnL detection by Alexa 488-conjugated antibody (red) (**2**), FITC-ABA lectin binding (green) (**3**), and a merging of both staining (**4**). The arrows indicate the superimposed detecting (yellow). Setae (white), acicula (blue), nerve cord (orange), and epidermis (gray) are shown. Scale bar: 300 μm (**B**).

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
