# Peer review of "Diverse Localization Patterns of an R-Type Lectin in Marine Annelids"

_molecules, 2021, doi:10.3390/molecules26164799_

Round 1
Reviewer 1 Report
The ms “ Diverse localization patterns of an R-type lectin in marine annelids” presented by Kawasar et al focuses on the cellular localisation of an R-type lectin PnL in a marine annelid.
The presented experimental work is straightforward and based on the generation of an antibody against PnL and probe it’s in vivo localization. Based on the different localization, potential sugar binding capacity of the lectin and a homology to pore forming toxins authors deduce possible a function of this type of lectin.
The work is interesting, but I have the feeling that it is rather unfinished as the real experimental data is descriptive and currently only leads to speculations that needs to be tested.
Altogether, it would elevate the level of this very interesting work with respect to industrial applications and the underlying the fundamental scientific knowledge if authors would include some experiments that confirm their speculations.
I suggest authors to setup and in vitro system where they used calcein-loaded vesicles and record, if present, the PnL-dependent membrane perforation measured by calcein release. See: Bavdek A, Gekara NO, Priselac D, Aguirre IG, Darji A, Chakraborty T, Macek P, Lakey JH, Weiss S, Anderluh G. Sterol and pH interdependence in the binding, oligomerization, and pore formation of Listeriolysin O. Biochemistry. 2007;46:4425–443.
Also of interest, as indicated by authors in the discussion, would be to include a glycoconjugate binding studies using a series of e.g. commercial Polyacrylamide-based glycoconjugates (Sigma Aldrich) to define the type of lectin that are important for the function of PnL and to acquire some solid data on the possible role of these type of R-Lectins to avoid general speculations.
The paper and its message would benefit from these type of experiments.
General Comments.
Section 2.1 belongs to Material and Methods as it is rather preparative and is the result of the optimization of a standard procedure, but this is not science, though it was when this type of procedures were not common.
Figure 1 can be included into Figure 2 or even be added as supplemental material.
Author Response
1. I suggest authors to setup and in vitro system where they used calcein-loaded vesicles and record, if present, the PnL-dependent membrane perforation measured by calcein release. See: Bavdek A, Gekara NO, Priselac D, Aguirre IG, Darji A, Chakraborty T, Macek P, Lakey JH, Weiss S, Anderluh G. Sterol and pH interdependence in the binding, oligomerization, and pore formation of Listeriolysin O. Biochemistry. 2007; 46:4425–443.
#Response: A calcein-release assay is interesting to perform. But unfortunately, we do not have that setup and our lab focuses mainly on protein-ligand interactions.
2. Also of interest, as indicated by authors in the discussion, would be to include a glycoconjugate binding studies using a series of e.g. commercial Polyacrylamide-based glycoconjugates (Sigma Aldrich) to define the type of lectin that are important for the function of PnL and to acquire some solid data on the possible role of these type of R-Lectins to avoid general speculations. The paper and its message would benefit from these types of experiments.
#Response: Although we can not prepare a new experiment on the polyacrylamide-based glycoconjugates during the short period until the revision, we added some possible biological functions of PnL based on your previous article (reference #10) in the discussion part at the lines 200-207.
General Comments.
3. Section 2.1 belongs to Material and Methods as it is rather preparative and is the result of the optimization of a standard procedure, but this is not science, though it was when this type of procedures were not common.
#Response: We changed the sub-title of section 2.1 from ‘preparation of antiserum’ to ‘Detection of PnL in the crude extract of lugworm tissues’ and modified the section to fit it in the ‘Results’.
4. Figure 1 can be included in Figure 2 or even be added as supplemental material.
#Response: We updated section 2.1. Previous Figure 1A moved to Supplemental Figure S1 (Now, the file is not attached because it is available to attach only one document) and Figure 1B has been included to the new figure 1A (that is Figure 2 mentioned above). Hope you will consider it.
Reviewer 2 Report
The authors detected various localization patterns of an R-type lectin in marine annelids. Generally speaking, the overall writing is good. Numerous data were generated to support the main conclusions. However, minor revisions are required before acceptance for publication.
- Provide the full name for each abbreviated term at its first appearance in the Abstract and main text.
- Extra editing is necessary. For example, Line 96: Where is the asterisk? Line 98: Change “black triangle” to “arrow”. Line 259: Change “previous reported” to “a previous report”. Line 264: Provide more information about Enago.
- Renumber the sub-figures in Figures 2, 3 and 4.
- Provide a paragraph to summarize the main conclusions.
Author Response
The authors detected various localization patterns of an R-type lectin in marine annelids. Generally speaking, the overall writing is good. Numerous data were generated to support the main conclusions. However, minor revisions are required before acceptance for publication.
1. Provide the full name for each abbreviated term at its first appearance in the Abstract and main text.
#Response: We provided the full form of PnL, Gal, GalNAc and SDS-PAGE in the abstract and methods.
2. Extra editing is necessary. For example, Line 96: Where is the asterisk? Line 98: Change “black triangle” to “arrow”. Line 259: Change “previous reported” to “a previous report”. Line 264: Provide more information about Enago.
#Response: Line 96: We removed the ‘asterisk’ from the figure legend as the arrow shows PnL in the crude extract. Line 98: We changed “black triangle” to “ blue arrow”. Line 259: We replaced “previous reported” by “a previous report”. Line 264: We provided the full information about Enago.
And, we performed the extra editing, such as:
(a) The figure legend of Fig. 3A is now modified.
(b) A scale bar is inserted in Fig. 3B. Perinereis nuntia is italicized in Line 62.
(c) Line 188: “……suggesting that they role innate immune molecules” is replaced with “……suggesting that they have roles as innate immune molecules”.
3. Renumber the sub-figures in Figures 2, 3 and 4.
#Response: We renumbered the sub-figures to "Figure 2 A1-4" from "Figure 2 Aa-d".
4. Provide a paragraph to summarize the main conclusions.
#Response: The last paragraph of the Discussion part is modified to summarize the main conclusions.
Round 2
Reviewer 1 Report
Authors performed requested modifications and though they did not include any of the suggested experiments due to technical and time limitations I have no more comments on this ms